# Piezo1 Mechano-Activation Is Augmented by Resveratrol and Differs between Colorectal Cancer Cells of Primary and Metastatic Origin

**DOI:** 10.3390/molecules27175430

**Published:** 2022-08-25

**Authors:** Joshua D. Greenlee, Kevin Liu, Maria Lopez-Cavestany, Michael R. King

**Affiliations:** Department of Biomedical Engineering, Vanderbilt University, PMB 351631, 2301 Vanderbilt Place, Nashville, TN 37235-1631, USA

**Keywords:** mechanotransduction, metastasis, shear stress, Piezo1, colorectal cancer, TRAIL, calcium, resveratrol

## Abstract

Cancer cells must survive aberrant fluid shear stress (FSS) in the circulation to metastasize. Herein, we investigate the role that FSS has on colorectal cancer cell apoptosis, proliferation, membrane damage, calcium influx, and therapeutic sensitization. We tested this using SW480 (primary tumor) and SW620 cells (lymph node metastasis) derived from the same patient. The cells were exposed to either shear pulses, modeling millisecond intervals of high FSS seen in regions of turbulent flow, or sustained shear to model average magnitudes experienced by circulating tumor cells. SW480 cells were significantly more sensitive to FSS-induced death than their metastatic counterparts. Shear pulses caused significant cell membrane damage, while constant shear decreased cell proliferation and increased the expression of CD133. To investigate the role of mechanosensitive ion channels, we treated cells with the Piezo1 agonist Yoda1, which increased intracellular calcium. Pretreatment with resveratrol further increased the calcium influx via the lipid-raft colocalization of Piezo1. However, minimal changes in apoptosis were observed due to calcium saturation, as predicted via a computational model of apoptosis. Furthermore, SW480 cells had increased levels of Piezo1, calcium influx, and TRAIL-mediated apoptosis compared to SW620 cells, highlighting differences in the mechano-activation of metastatic cells, which may be a necessary element for successful dissemination in vivo.

## 1. Introduction

Colorectal cancer (CRC) remains the second leading cause of cancer-related death in the United States [1]. As is true for most cancers, the primary cause of CRC-related death is attributed to metastasis [2]. Metastasis is the process in which cancer cells undergo an epithelial to mesenchymal transition (EMT) to become increasingly motile, which then allows them to migrate toward the vasculature and then intravasate into vessels through the endothelial junctions. For successful tumor dissemination to occur, once cancer cells access blood or lymphatic vessels, they must survive inhospitable environments comprised of surveying immune cells and harsh physical forces from flow. For example, in lymphatic vessels, cancer cells have been shown to experience pulses of shear stress ranging from 4–12 dyne/cm^2^ compared to less than 0.1 dyne/cm^2^ within a solid tumor [3,4]. Within the blood, cancer cells experience magnitudes of circulatory shear stress ranging from 0.5–30 dyne/cm^2^ within veins and arteries to magnitudes greater than 1000 dyne/cm^2^ for brief periods of time at arterial bifurcations and within the heart [5,6]. These harsh fluid environments explain why less than 0.01% of circulating tumor cells (CTCs) that enter the circulation survive to colonize secondary sites in the body [7,8]. However, the presence and survival of CTCs is a strong predictor of secondary-tumor colonization, poor prognosis, and mortality in most cancer types [9,10,11].

Our laboratory has recently demonstrated that prostate cancer cell lines have different sensitivities to fluid shear stress (FSS) that are dependent upon their metastatic location [12]. Cells derived from a brain metastasis (having been exposed to high magnitudes of FSS from the heart) were much more resistant to FSS-induced apoptosis than cells derived from the lymph node. However, these cell lines were isolated from different patients, making it more difficult to interpret the mechanisms of FSS resistance. Using an isogenic cellular model consisting of cell lines derived from different tissues within the same patient could provide mechanistic insight into how cancer cells evolve to evade apoptosis within the circulation. Recent studies have shown that physiological levels of FSS stress can promote a mesenchymal phenotype and enhance CTC survival in the circulation [13,14,15,16]. For example, one study demonstrated that lung cancer cells exposed to fluid shear stress show side-population enrichment, hallmarks of EMT, and upregulation of cancer stem cell (CSC) marker CD44 [16]. However, no study has compared the response to FSS between cells isolated from the primary tumor with those that have successfully disseminated to a metastatic site. We hypothesize that cancer cells of metastatic origin will be increasingly impervious to FSS through mechanisms of intrinsic (subpopulation selection) or acquired resistance.

Cancer cells are able to sense and respond to FSS and other mechanical stimuli through the process of mechanotransduction. One type is the stimuli-induced opening of mechanosensitive ion channels that allows for the influx of calcium, which acts as a secondary effector to propagate intracellular signaling cascades. Calcium-gated ion channels, such as transient receptor potential cation channel subfamily V member 4 (TRPV4), P2X purinoceptor 7 (P2X7), and Piezo1, have recently been implicated as integral components in mechanisms of cancer metastasis and cell death [17,18]. For instance, we have recently determined that cancer cells can be sensitized to tumor necrosis factor-related apoptosis inducing ligand (TRAIL)-mediated apoptosis through the activation of Piezo1, either through FSS or chemical agonists such as Yoda1 [19,20]. There is evidence suggesting that both the lipid composition and biophysical properties of the membrane influence mechanosensitive ion-channel activation [21,22]. The concentration of membrane cholesterol has effects on cell stiffness, tension, and rigidity, consequently affecting membrane deformation and the force “perceived” by channels following mechanical stimuli. For example, the depletion or disruption of membrane cholesterol has been shown to inhibit the activation of mechanosensitive ion channels, including Piezo1 [23,24]. If the cell membrane can be considered a spring, increasing the membrane rigidity via cholesterol (a higher spring constant) will make the applied force greater for equivalent membrane deformations.

Cholesterol is an integral component of lipid rafts (LRs), which are densely packed, organized, detergent-insoluble areas within the lipid bilayer [25,26]. Lipid rafts have been shown to facilitate the mechanisms of cancer metastasis by acting as protein scaffolds and receptor oligomerization platforms to augment downstream signal transduction [27]. While the role of LRs in many forms of signal transduction, including apoptotic death receptor signaling, has been well described [28], the role of rafts in mechanosensation via ion channels remains elusive. We hypothesize that, similar to cholesterol, the presence of lipid rafts will have an amplifying effect on Piezo1 activation and calcium influx. In this study, we examine the role that lipid rafts and the raft-stabilizing polyphenol resveratrol (RSV) have on the mechanical and chemical activation of Piezo1 in order to sensitize cancer cells to therapeutics. Additionally, using two CRC cell lines (SW480 and SW620) derived from the same patient, we demonstrate that metastatic cells are increasingly resistant to different types of FSS. Understanding how cells respond to different types of physiological FSS and mechanosensitive ion-channel agonists may prove useful in leveraging calcium-dependent metastatic processes for therapeutic benefit.

## 2. Results

### 2.1. SW480 Cells from the Primary Tumor Are Increasingly Sensitive to FSS

For this study, two CRC cell lines derived from the same patient were used. SW480 cells were isolated from a primary adenocarcinoma of the colon in a 50-year-old male, while SW620 cells were isolated from a lymph node metastasis one year later. Through metastasizing from lymphatic vessels into a lymph node, the SW620 cells would have been exposed to higher magnitudes of FSS than the SW480 cells in the body. SW620 cells have been well characterized as being increasingly mesenchymal, tumorigenic, and metastatic compared to SW480 cells, which appear more epithelial in culture [29]. However, no study to date has compared how these cell lines respond to FSS. To measure the sensitivity of these cell lines to FSS, cells were exposed to three different physiological flow conditions: static (0 dyne/cm^2^), shear pulses (10 pulses of 3950 dyne/cm^2^; total strain = 39.5 dyne × s/cm^2^), and constant shear (2.5 h of 10 dyne/cm^2^; total strain = 90,000 dyne × s/cm^2^) (Figure 1). Shear pulses were used to model brief intervals of high-intensity FSS in areas such as the heart, whereas constant shear was used to model average magnitudes and durations that CTCs experience in circulation [12,30,31]. After FSS, the cells were incubated for 24 h and then analyzed for apoptosis via Annexin V/propidium iodide (PI) staining. The SW480 cells had a significant decrease in cell viability from both forms of FSS compared to SW620 cells (Figure 2A,E). Decreases in viability coincided with increases in the percentage of cells undergoing early and late-stage apoptosis (Figure 2B,C). Metastatic SW620 cells demonstrated no significant change in cell viability in response to the FSS. To determine if the FSS-induced apoptosis was associated with enhanced mitochondrial outer membrane permeability (MOMP), the cells were stained with a JC-1 dye after FSS. Both the SW480 and SW620 cells showed no changes in depolarized mitochondria following FSS, suggesting that increases in apoptosis were likely due to the extrinsic apoptotic pathway (Figure 2D).

### 2.2. Moderate Levels of Constant FSS Decrease Cell Proliferation and Increase CD133 Expression

To measure changes in proliferation and the cell cycle after FSS, the cells were analyzed for Ki67 expression and total DNA content, respectively. Interestingly, cell proliferation was significantly decreased after constant shear stress but not shear pulses in SW480 cells (Figure 3A). SW620 cells also saw a 20% decrease in proliferation from constant shear, but this was not found to be significant. Despite decreases in proliferation, there was no significant change in the cell cycle (Figure 3B). Studies have shown that exposing suspended cancer cells to fluid flow can alter the expression of mesenchymal and CSC markers [16]. Twenty-four hours post-FSS, the cells were analyzed for CSC markers CD133 and CD44. To exclude dead cells, the samples were counterstained with propidium iodide before analysis, where debris and propidium iodide (+) cells were excluded from analysis (Figure 3C). The SW480 cells were found to be high expressors of CD44 but low in terms of CD133, whereas the SW620 cells had low CD44 expression and high CD133. While the CD44 expression remained unchanged from FSS (Figure 3D), constant FSS increased the expression of CD133+ SW480 cells by over 2-fold (Figure 3E). CD133 is a transmembrane protein and CSC marker in many cancer types, and it is a predictor of cancer-cell survival, tumorigenicity, and metastasis [32]. This increase in CD133 expression in SW480 cells more closely mimics the expression in metastatic SW620 cells, possibly highlighting a mechanism by which cells in transit become more metastatic though their response to sustained fluid forces.

### 2.3. High-Magnitude FSS Pulses Cause Cell-Membrane Damage and Pores That Are Rapidly Repaired

We previously demonstrated that pulses of high-magnitude FSS cause membrane damage in prostate cancer cells [12]. To investigate the role of the cell membrane in relation to apoptosis during FSS, cells were incubated with fluorescently-conjugated dextran of 3kD and 10kD MW during FSS exposure. These 3kD and 10kD MW dextran molecules have an estimated hydrodynamic radius of 1.4 nm and 2.3 nm, respectively. Cells with an uptake of fluorescent dextran were considered damaged from the formation of membrane pores that were greater in size than the dextran hydrodynamic radius. Interestingly, shear pulses caused a dramatic increase in damaged cells (over 6-fold in both SW480 and SW620 cells) as indicated by the internalization of the 3kD MW dextran (Figure 4B). Thfe pulses also caused an increase in pore formation exceeding 2.3 nm in size, which was most pronounced in the SW480 cells with a significant increase of 3-fold (Figure 4C). This membrane damage was unique to high-magnitude shear pulses, as constant shear stress did not cause any appreciable change in dextran internalization. To measure cell-membrane recovery and repair, cells were incubated with PI 20 min after FSS. Cells that were positive for PI were considered “unrecovered”, as they were unable to sufficiently repair pores in the membrane and were therefore permeable to PI (Figure 4A). Despite the significant increase in cell-membrane damage from shear pulses, over 80% of SW480 and SW620 cells were able to repair their membranes within minutes of FSS-induced damage (Figure 4B,C). This rapid membrane recovery is thought to explain why both the SW480 and SW620 cells were relatively resistant to apoptosis from shear pulses.

### 2.4. FSS and Resveratrol Sensitize SW480 Cells to TRAIL-Mediated Apoptosis

Despite significant decreases in cell viability in the SW480 cell line after FSS, both cell lines remained relatively resistant, and less than 30% of cells were apoptotic. Circulatory shear stress has been shown to sensitize cancer cells to chemotherapeutic agents and targeted therapies. We demonstrated this previously in multiple cancer types with the use of tumor necrosis factor alpha-related apoptosis-inducing ligand (TRAIL) [20]. Shear pulses significantly sensitized SW480 cells to 50 ng/mL of TRAIL, while constant shear had minimal effects (Figure 5A,B). Interestingly, both forms of shear stress had no effect on TRAIL sensitization in SW620 cells (Figure 5C).

To investigate whether these cancer cells could be further sensitized to TRAIL via lipid raft alteration, we added a condition where cells were pretreated with resveratrol before FSS. Resveratrol (RSV) is a polyphenol compound that has been shown to have pleotropic effects on cancer cells, especially through its ability to form tightly packed liquid-ordered domains in the cell membrane [33]. Resveratrol has the ability to sensitize cancer cells to TRAIL under static conditions but has not been investigated in the context of physiological FSS [28,34]. Additionally, resveratrol predominantly integrates into the plasma membrane within minutes of treating cells [35]. Taking this into account, the cells were pretreated with 50 µM resveratrol for just 1 h, washed to remove trace amounts in solution, and then immediately exposed to FSS treatments to restrict the effects of resveratrol to the cell membrane. The SW620 cells were significantly sensitized to TRAIL when pretreated with resveratrol under static conditions. However, the cell viability remained similar between static and shear-treated cells regardless of treatment, further demonstrating the mechanoresistant phenotype of these cells (Figure 5C). In SW480 cells, the shear pulses significantly reduced viability in the TRAIL + RSV treated condition, but this was not significantly different from the FSS and TRAIL alone (Figure 5B). While FSS and resveratrol sensitize these cells to TRAIL individually, there were no appreciable synergistic affects when combined.

Our lab determined that the mechanism for FSS-mediated TRAIL sensitization is through calcium influx [19]. To measure calcium influx in real time, the cells were incubated with the acetoxymethyl (AM) ester calcium dyes Fluo-4 and FuraR, exposed to one pulse of shear stress, and then immediately analyzed via flow cytometry (Figure 5D). Interestingly, there were minimal changes to the intracellular calcium concentrations between the static and shear conditions for both the SW480 and SW620 cells (Figure 5E,F). Resveratrol caused slight, albeit insignificant, increases in calcium influx, potentially explaining the minimal changes in cell viability. While mechanical stimulation had few effects on the intracellular calcium concentration and TRAIL-mediated apoptosis, more substantial calcium influxes were observed through treatment with chemical agonists of mechanosensitive ion channels [19].

### 2.5. Resveratrol Increases Calcium Influx Induced by Yoda1 Chemical Activation of Piezo1

To further examine the effects of mechanosensation in our isogenic CRC cell lines, we selectively activated Piezo1, a mechanosensitive ion channel that has recently been implicated in the mechanism of calcium entry and TRAIL sensitization [19]. Yoda1 is an agonist of the calcium-gated mechanosensitive ion channel Piezo1, acting as a molecular wedge within the transmembrane domain. While pretreatment with resveratrol had few effects on calcium influx from FSS, resveratrol significantly increased the calcium concentrations after 24 h of 10 µM Yoda1 treatment (Figure 6A,B). Interestingly, resveratrol treatment also caused the cells to appear more mesenchymal in shape compared to Yoda1-only-treated cells. The cell aspect ratio (a metric of spindle-shaped morphology and hallmark of EMT) was increased in resveratrol-treated cells and trended in a similar manner to calcium influx (Figure 6F). To measure instantaneous changes in cytosolic calcium, the cells were treated as previously described; however, the pulse of FSS was substituted with a bolus addition of Yoda1 immediately preceding analysis. Resveratrol-pretreated cells showed significantly increased activation of Piezo1, as indicated by the increased concentrations of intracellular calcium (Figure 6C,E). The calcium influx increased by 2-fold immediately following Yoda1 treatment and remained so for the 200 sec duration of measurement (Figure 6D).

### 2.6. SW480 Cells Are Increasingly Mechanosensitive Compared to SW620 Cells

While SW480 cells showed a significant increase in calcium influx from the Yoda1–resveratrol combination treatment, there was no measurable difference for the SW620 cells (Appendix A). Additionally, the total calcium influx remained significantly lower for SW620 cells compared to SW480 cells for all Yoda1-treated conditions (Figure 6F). This, in combination with our earlier data showing FSS-induced apoptosis and TRAIL sensitization, further supports the enhanced mechanosensation of SW480 cells compared to their SW620 counterparts. Notably, the levels of calcium from Yoda1 treatment were approximately an order of magnitude higher than those observed after shear pulses. Considering that the level of cytosolic calcium remained relatively unchanged, and the fact that we previously showed that cell membrane damage is rapidly repaired, this could be explained by calcium-mediated mechanisms of lysosomal exocytosis, membrane patching, and cytosolic calcium buffering [36,37,38,39]. Membrane damage causes increased cytosolic calcium concentrations that remain local to the damaged site due to cytosolic buffering mechanisms, which can decrease calcium changes by orders of magnitude in just milliseconds [36,40,41]. However, Yoda1 has been shown to initiate the sustained activation of Piezo1 through the stabilization of the open state for longer intervals [42].

### 2.7. Resveratrol Increases Colocalization of Piezo1 with Lipid Rafts

Numerous studies have shown that lipid rafts can amplify downstream-signaling cascades through receptor oligomerization, supramolecular clustering, and the scaffolding of membrane proteins [43,44]. However, little is known regarding lipid raft-mechanosensitive ion channel interplay, and how this affects channel activation. Staining for lipid rafts and Piezo1 before and after resveratrol treatment, we found that resveratrol not only increased lipid-raft presence, but also increased the incidence of Piezo1 in these raft domains (Figure 7A). Resveratrol significantly increased the Pearson’s correlation coefficient between Piezo1/LR in both the SW480 and SW620 cells (Figure 7B). Further, resveratrol increased the FRET efficiency of Piezo1 (donor) and LR (acceptor) in both cell lines (Appendix A). The SW480 cells were found to have significantly increased expression of Piezo1 compared to the SW620 cells (Figure 7C,D). This is supported by the protein abundance data publicly available from the Broad Cancer Cell Line Encyclopedia [45]. Of all cell lines with proteomic data, there were two sets isolated from the same patient: SW480 and SW620 cells and WM115 and WM2664 cells (melanoma cells isolated from the primary tumor and metastatic lymph node, respectively). Similar to our colon adenocarcinoma cell lines, the metastatic melanoma cell line exhibited a decreased abundance of Piezo1 compared to cells from the primary tumor (Figure 7E). While this is only shown for two same-patient cell line models, this suggests a pivotal mechanism of Piezo1 downregulation for successful survival in the circulation. FSS treatments had no effect on Piezo1 expression in either cell line (Appendix A).

### 2.8. Resveratrol-Induced Increase in Piezo1 Activation Has Minimal Effect on Apoptosis Due to Calcium Saturation

We hypothesized that the substantial changes in calcium influx from resveratrol + Yoda1-treated SW480 cells would have a significant effect on TRAIL sensitivity. Surprisingly, the addition of resveratrol had no significant effect on viability following the Yoda1 and TRAIL combination treatment (Figure 8A). Yoda1 sensitization, the percent decrease in viability from the addition of 10 µM Yoda1, remained unchanged between TRAIL and TRAIL + resveratrol treated conditions (Figure 8B). To better understand this unexpected result, a computational model was adapted from our previous work to study how changes in calcium amplify the effects of the TRAIL treatment and cause increased cellular apoptosis [19]. Apoptosis was simulated over a 24 h time period by calculating the cleaved PARP (cPARP) concentration within the cells (Appendix A). Values over 5 × 10^5^ were considered above the threshold indicative of apoptosis. Additionally, MOMP was calculated via the concentration of cytosolic Smac. The simulation was run across a range of different TRAIL concentrations, 0.5, 50, and 200 ng/mL, with 50 ng/mL matching the experiments performed in vitro. Calcium concentrations were taken from the flow cytometry median ratiometric fluorescence values (Figure 6E) and normalized to make the Yoda1 calcium concentration equal to 1 µM. This yielded cytosolic free-calcium concentrations of just over 100 nM for control conditions (Appendix A), consistent with average concentrations at a resting state [46].

At the 50 ng/mL TRAIL concentration, the calcium-free, control, and resveratrol treatment conditions showed that the cPARP concentration did not reach a high enough level to be considered apoptotic (Figure 8E). Yoda1 calcium concentrations resulted in the cPARP concentration reaching the apoptotic threshold at *t* = 15 h. Interestingly, at this TRAIL concentration, the increase in calcium caused by treatment with both Yoda1 and resveratrol did not result in a change in cPARP behavior in the computational model compared to just Yoda1 treatment. MOMP occurred at *t* = 11 h in both the Yoda1 and the Yoda1–resveratrol treatment conditions (Figure 8F). The peak cytosolic Smac concentration, on average, was reached 4 h later in the simulations with lower intracellular calcium concentrations. The model results at the 0.5 ng/mL TRAIL condition were similar to those at the higher TRAIL dosage. In the Yoda1 and Yoda1 treatment conditions, no differences were calculated in cell apoptosis or MOMP times. The apoptotic threshold for cPARP was at the 19 h timepoint (Figure 8C). This is 4 h later than the simulations corresponding to 50 ng/mL of TRAIL as an initial condition. The maximum concentration of cytosolic Smac for both groups occurred at the 17 h timepoint (Figure 8D). In the groups with lower initial calcium concentrations, the MOMP occurred 5 h later than the high calcium concentration conditions. Treatment with 200 ng/mL of TRAIL in the computational model showed minimal differences from the results using 50 ng/mLin both the cPARP and cytosolic Smac concentrations (Figure 8G,H). These results parallel the trends seen in vitro, where increasing the TRAIL concentrations from 50 to 200 ng/mL had little effect on cell viability (Appendix A). According to these simulations, and in agreement with our experimental results, the amount of calcium needed to sensitize the cells to TRAIL was saturated at 1 µM by just Yoda1 treatment. Therefore, the additional influx of calcium (1.7 µM) by also treating with resveratrol had little effect on cell viability.

In order to better understand how variation in cancer cell populations can affect TRAIL sensitization via increased intracellular calcium concentrations, heterogeneous cell populations were simulated by running the model with randomly distributed cytosolic Bcl-2 and XIAP concentrations. These proteins are inhibitors of apoptosis and were chosen based on their importance in regulating MOMP [19]. Simulated cell populations were treated with each of the calcium concentrations described throughout the study to assess the changes in cell viability in each randomized cell population. Overall, the model results show consistency with the experimental results. Figure 9A shows the model output for cPARP over the course of 24 h in the calcium-free condition treated with 50 ng/mL of TRAIL, where each individual line is the fate of one of the 1000 cells in the randomized population. The simulated cell viability in the calcium-free condition was 71%. The viability was similar to the TRAIL treatment in the control (Figure 9B) and the resveratrol only (Figure 9C) conditions, calculated to be 68% and 70%, respectively. In the groups with increased intracellular calcium concentrations, the effect of TRAIL on cancer cell viability was more pronounced. Yoda1 and TRAIL treatment resulted in 47% cell viability after 24 h in the simulated experiments (Figure 9D). Adding resveratrol had a minimal effect on the sensitization of the cancer cells to TRAIL, with a calculated cell viability of 46%. (Figure 9E). The in vitro and in silico models of calcium influx maintained high degrees of fidelity, confirming the saturating effect of calcium concentrations on cellular apoptosis above 1 µM (Figure 9F).

## 3. Discussion

Previous studies have shown that cancer cells are more resistant to fluid shear stress than cells from healthy epithelial origins [15,47,48]. Additionally, there is evidence that the propensity for FSS survival may predict metastatic organotropism in prostate cancer cell lines [12]. Similarly, our results demonstrate that CRC cells derived from a metastatic lymph node are more resistant to FSS than cells from the primary tumor. Given the isogenic nature of our cell lines, we can, therefore, draw conclusions on the phenotypic changes these cells undergo in order to survive and adapt to different fluid environments. While cells from the primary tumor showed the most significant increases in apoptosis, both cell lines remained relatively resistant to FSS death and displayed modest changes in mitochondrial depolarization. Moving forward, it is imperative that we work to understand the mechanisms of shear resistance in metastatic cells. Similarly to mechanisms of chemoresistance, which can be either intrinsic or acquired, future studies should examine whether FSS selects for subpopulations of cancer cells that are intrinsically insensitive to these physical forces, or if cells are primed by the circulation to make them more “mechanoresistant”.

Our results show that not all forms of fluid shear stress are alike. While both forms of FSS caused decreases in SW480 cell viability, cells responded in very distinct ways across these conditions. Shear pulses of high magnitudes acted in a manner dependent upon the biophysical properties of the cell membrane. Despite significant increases in cell membrane damage and the formation of pores at least 2.3 nm in size, the cells were able to rapidly repair their membranes, likely explaining their ability to resist death. We demonstrated that cell stiffness strongly correlates with FSS survival, and softening agents that disrupt actin polymerization, such as cytochalasin D, can increase sensitivity to FSS [12]. However, SW480 cells have been shown to have a higher Young’s modulus and stiffer cytoskeleton than SW620 cells [49], suggesting that cell stiffness alone may not be an accurate predictor of FSS survival. For instance, the endocytosis of plasma membrane wounds and the subsequent repair mechanisms have been shown to occur in both a calcium- and lipid-raft-dependent manner [36,50].

Meanwhile, shear stress treatments mimicking the conditions experienced by CTCs in the circulation caused decreased proliferation, most pronounced in SW480 cells. This is consistent with another study of CRC cells, where FSS decreased proliferation but increased expression of β-catenin and c-myc [51]. Furthermore, c-myc has been implicated not only as a regulator of proliferation, but also in the expression of CSC markers such as CD133 in gliomas and other cancers [52]. This is consistent with the present results, as a constant shear stress of 10 dyne/cm^2^ caused the significant upregulation of CD133 expression in surviving SW480 cells. This increase in CSC populations with EMT hallmarks and increased metastatic potential has also been observed in breast and lung cancers [13,14]. Additionally, decreased proliferation and increased invasiveness are hallmarks of the CSC phenotype [53]. Our results further support the role of FSS in metastatic priming.

These data also suggest that prolonged FSS may desensitize metastatic cancer cells to mechanisms of mechanotransduction. Metastatic SW620 cells were not only more resistant to apoptosis, but also resisted FSS-induced TRAIL sensitization and calcium influx upon Piezo1 activation. Furthermore, examining the protein abundance data in primary and metastatic cells derived from the same CRC and melanoma patient, we highlighted that cells derived from lymph-node metastases showed consistently downregulated expressions of Piezo1. This is consistent with our in vitro results, which show that surface expression of Piezo1 was decreased in SW620 cells. This begs the question of whether the downregulation of mechanosensitive ion channels such as Piezo1 is necessary for metastasis to occur. Another possibility is that FSS naturally selects for subpopulations of cancer cells that are impervious to the stress of shear due to lower innate levels of Piezo1. This is an interesting dichotomy, as numerous studies have implicated Piezo1 as a driver of cancer progression and many pro-metastatic processes [18]. Future studies should elucidate this role of Piezo1 in priming CTCs for survival in transit, for instance by using additional isogenic cell models of metastasis.

There is strong evidence that plasma membrane lipid composition has an effect on mechanosensitive ion channels such as Piezo1 [23,24]. In prior studies, the depletion of cholesterol from the cell membrane using methyl-β-cyclodextrin decreased the chances of channel activation from mechanical stimuli and increased the force needed for opening. This highlights the biophysical requirements of the cell membrane for proper mechanosensitive ion-channel activation, as cholesterol has been shown to increase membrane stiffness and rigidity [54]. Cholesterol is also an integral component of lipid rafts; however, surprisingly, there are no studies investigating the role of lipid rafts in mechanosensitive ion channel function. Treatment with resveratrol, which has been well characterized to rapidly assimilate into the cell membrane to promote the formation of lipid-ordered domains [33,35], significantly increased Piezo1 colocalization with lipid rafts in both cell types. However, this increased raft colocalization corresponded to increased calcium influx and Piezo1 activation selectively in SW480 cells. This is hypothesized to be due to the aforementioned higher Piezo1 content in these cells. We postulate that, similar to cholesterol, lipid rafts and resveratrol increase membrane rigidity locally, thereby stabilizing Piezo1 in an open state once activated via mechanical deformation or molecular wedging. Future studies should examine the molecular and biophysical mechanisms of this phenomenon.

Despite resveratrol’s ability to facilitate the activation of Piezo1, we observed that ultimately there was little resulting enhancement to cell apoptosis. This was confirmed in our computational model of apoptosis, where cytosolic calcium concentrations exceeding 1 µM had few effects on cleaved PARP, the release of cytosolic Smac, and eventual apoptosis. Despite a minimal change in apoptosis, it is possible that this increase in calcium could have consequential effects on other cellular processes not directly examined here. Calcium is a well-studied secondary messenger in many oncogenic and metastatic signaling processes, such as migration, invasion, proliferation, and EMT [55]. In this study, we observed that increased calcium concentrations from Yoda1 and resveratrol correlated with an increased aspect ratio, a hallmark of EMT. While there were insignificant changes in apoptosis within our in vitro and in silico models, future studies should examine these other calcium-mediated downstream effects related to cancer progression and metastasis.

## 4. Conclusions

SW480 cells originating from a primary tumor are more sensitive to FSS than SW620 cells that were isolated from a metastatic site in the same patient. The SW480 cells showed enhanced apoptosis, senescence, calcium influx, and TRAIL sensitization following FSS. High-magnitude FSS pulses caused the formation of pores in the cell membrane, which were rapidly repaired. Meanwhile, sustained levels of moderate FSS induced stem-like features, including increased CD133 expression and decreased cell proliferation. Metastatic SW620 cells exhibited decreased Piezo1 expression, suggestive of Piezo1 downregulation for the survival of cancer cells in transit. Future studies should examine this clinically to see if non-apoptotic CTCs and cells from metastatic lesions are low expressors of Piezo1, or if this effect is unique to the circulation. This may inform treatment strategies for the small subpopulations of CTCs that are able to survive and colonize secondary sites. Additionally, the Piezo1 molecular agonist Yoda1 significantly increased intracellular calcium levels in SW480 cells, which was further increased through resveratrol-induced lipid raft colocalization of Piezo1. This presents a novel role for lipid rafts in the enhanced activation of Piezo1, which may also be present in other mechanosensitive ion channels. Moving forward, the clinical utilization of lipid-raft-altering agents such as resveratrol or methyl-β-cyclodextrin should be explored to enhance or curb Piezo1 activation, respectively. Calcium is a secondary messenger that is involved in many premetastatic processes. Indirectly tailoring Piezo1 activation via pharmacological alteration of lipid rafts may be an elegant way to curb mechanotransduction events essential to metastasis. However, improved systemic delivery strategies will be crucial for translational adoption into clinics.

## 5. Materials and Methods

### 5.1. Cell Culture

Colorectal cancer cell lines SW620 (ATCC, #CCL-227) and SW480 (ATCC, #CCL-228) were purchased from American Type Culture Collection. Cells were cultured in RPMI1640 cell culture medium (Gibco, Grand Island, NY, USA). Medium was supplemented with 10% (*v*/*v*) fetal bovine serum and 1% (*v*/*v*) PenStrep, all purchased from Invitrogen. Cells were passaged at 70% confluency by washing with HBSS without calcium or magnesium (Gibco) and lifting with Trypsin-EDTA (0.25%) (Invitrogen, Grand Island, NY, USA). Both cell lines were maintained in a humidified incubation chamber at 37 °C and 5% CO_2_. SW480 cells are characterized as being epithelial in morphology with apical–basal polarity, while SW620 cells are increasingly mesenchymal with a loss in epithelial characteristics and polarity. Cells were screened for mycoplasma contamination and tested negative.

### 5.2. Fluid Shear Stress Treatments

To model different types of physiological shear stress, the cells were exposed to three different shear conditions: static, shear pulses, or constant shear. For shear pulses, 2 mL of media containing 200,000 cells/mL was loaded into a 5 mL syringe with a 30-gauge x ½ in needle. Syringes and needles were blocked for 30 min with 5% bovine serum albumin (BSA, Sigma, St. Louis, MO, USA) in HBSS before loading. Cells were sheared using a syringe pump (Harvard Apparatus) at a flow rate of 14 mL/min, as previously described [12]. The average wall shear stress was calculated to be 3950 dyne/cm^2^ (395 Pa). Cells were exposed to 10 pulses with 2 min between each pulse, modeling the time it would take a cell to pass once through the entire circulatory system circuit [30]. To model the constant fluid shear stress, 0.5 mL of cells in media was loaded into a cone-and-plate viscometer (Brookfield LVDVII) at a concentration of 1 million cells/mL (higher cell concentrations were used to increase viscosity and subsequently increase shear stress). Before loading the cells, both the spindle and cup were blocked with 5% BSA for 30 min. Cells were sheared at 100 RPM (740 s^−1^) using the CP-40Z spindle for 2.5 h with an approximate shear stress of 10 dyne/cm^2^. Following treatment, the cells were collected, and the cup and spindle were rinsed with 0.5 mL of fresh media to collect remaining cells. Static cells were left in a 1.5 mL conical tube at a concentration of 200,000 cells/mL while shear experiments were being performed. After FSS treatments, 1 mL of cell suspension was plated in a 12-well plate (CELLTREAT) and incubated overnight for subsequent analysis.

### 5.3. Annexin V/Propidium Iodide Apoptosis Assay

Cells were incubated for 24 h after shear treatments, then collected by recovering the supernatant and lifting the remaining adhered cells using 0.25% Trypsin-EDTA (Gibco). Cells were washed thoroughly with HBSS with calcium and magnesium and incubated for 15 min with FITC-conjugated Annexin-V and propidium iodide (PI) (BD Pharmingen) at room temperature (RT) in the absence of light. Cells were immediately analyzed using a Guava easyCyte 12HT benchtop flow cytometer (Millipore Sigma). Viable cells were identified as negative for both Annexin-V and PI, early apoptotic cells were positive for Annexin-V only, late apoptotic cells were positive for both Annexin-V and PI, and necrotic cells were positive for PI only. Flow cytometry plots were analyzed using FlowJo v10.8.0 software. Control samples included: unstained negative control with no Annexin-V/PI to adjust for background and autofluorescence and Annexin-V-only and PI-only samples for gating and compensation purposes.

### 5.4. JC-1 Mitochondrial Membrane Potential Assay

Following FSS treatments and overnight incubation, the cells were lifted and collected as previously described. Cells were washed with HBSS without calcium or magnesium and incubated for 15 min with JC-1 dye (Abcam, Cambridge, UK) in a humidified incubation chamber at 37 °C. Cells were washed, and JC-1 fluorescence was assessed via flow cytometry. Cells with healthy mitochondria were identified as having a higher red fluorescence and a lower green fluorescence, while those with depolarized mitochondria showed a lower red JC-1 fluorescence and a higher green fluorescence.

### 5.5. Cell Cycle and Ki67 Proliferation Assay

Cells were incubated for 24 h after shear treatments, then collected by recovering the supernatant and lifting the remaining adhered cells using 0.25% Trypsin-EDTA (Gibco). Cells were washed twice with HBSS with calcium and magnesium, then resuspended in 50 µL of HBSS −/−. Then, 450 µL of chilled 70% ethanol in DI water was added to the cell suspension in a dropwise manner while vortexing to minimize cell aggregation. The cell suspension was incubated for 2 h at −20 °C. Cells were washed twice with FACS buffer (HBSS with 1% BSA), then incubated with 5 µL PE anti-human Ki-67 (BioLegend, San Diego, CA, USA, clone Ki-67) in 100 µL HBSS for 30 min at RT. Cells were washed twice, then incubated with 100 µL DAPI solution (3 µM in FACS buffer) for 15 min. Samples were then analyzed via flow cytometry using the Blue-V (DAPI) and Yellow-B (PE-Ki67) channels. Cell cycle was determined using Watson Pragmatic modeling in FlowJo. Cell proliferation was determined by assessing the Ki67+ cell population, and gates were established from an unstained control sample.

### 5.6. Cell Membrane Damage and Repair

Cells were sheared as previously described in the presence of 10 µM FITC-conjugated dextran of 3000 or 10,000 MW (Invitrogen). Cells were incubated for 5 min before shear and maintained in the dextran solution for the duration of the shear treatment. The cells were then allowed to recover for 20 min post shear, washed with HBSS with calcium and magnesium, and incubated with PI (hydrodynamic radius ~0.8 nm) for 10 min to measure membrane repair. Samples were analyzed via flow cytometry using Green-B (FITC) and Red-B (PI) channels. Cells positive for FITC-dextran but negative for PI were considered damaged and repaired, while FITC-dextran positive/PI positive cells were interpreted to be damaged and unrepaired.

### 5.7. Cancer Stem Cell Markers

After FSS treatments, the cells were collected by recovering the supernatant and lifting the remaining adherent cells using 0.25% Trypsin-EDTA. Cells were fixed in 4% PFA in HBSS for 15 min at RT, washed twice with HBSS, then blocked in 100 μL FACS buffer for 30 min at 4 °C. The samples were stained with 5 μL of both Brilliant Violet 421 anti-human CD133 (BioLegend, clone 7) and FITC anti-human CD44 (BioLegend, clone BJ18) for 30 min at 4 °C. The samples were washed twice with HBSS and then incubated with PI to exclude dead cells from the analysis. Samples were analyzed using a Guava easyCyte flow cytometer utilizing Blue-V (Brilliant Violet 421), Green-B (FITC) and Red-B (PI) channels.

### 5.8. TRAIL and Resveratrol Treatments

A 20 mM stock solution of resveratrol (Sigma-Aldrich) in DMSO was made fresh before each experiment. After cells were collected, but prior to shear/Yoda1 treatments, cells were incubated with 50 µM resveratrol for 1 h at 37 °C in serum-free media. To remove trace amounts of resveratrol remaining in solution that were not taken up by the cells, the samples were centrifuged at 300× *g* for 5 min, and the medium was aspirated. Cells were resuspended in serum-containing media in the presence or absence of 50 ng/mL TRAIL. For FSS experiments, cells were then sheared as described previously and plated in 12-well cell culture plates for 24 h. For Yoda1 experiments, cells were treated with 10 µM Yoda1 (Tocris) and then incubated for 24 h. The next day, apoptosis was analyzed using Annexin V/PI staining and flow cytometry. Yoda1 sensitization was calculated using the following formula:$$\mathrm{Yoda}1\;\mathrm{Sensitization}=\frac{\left(\%\mathrm{DMSO}\;\mathrm{viability}\right)-\left(\%\mathrm{Yoda}1\;\mathrm{viability}\right)}{\left(\%\mathrm{DMSO}\;\mathrm{viability}\right)}*100\%$$
for each treatment condition (Control, TRAIL, resveratrol, and TRAIL+ resveratrol).

### 5.9. Calcium Influx

A total of 200,000 cells was collected in 1 ml of serum-free media and incubated for 30 min with 1 µM Fluo-4 and 2 µM Fura Red (Invitrogen) at 37 °C. For resveratrol-treated samples, cells were treated for 1 h with 50 µM resveratrol before being incubated with calcium dyes. The cells were washed in calcium/magnesium-free HBSS, then resuspended in HBSS with calcium and magnesium for 30 min at RT in the dark. For FSS samples, the cells were sheared using the syringe-pump system for 1 pulse, then immediately analyzed via flow cytometry. For Yoda1 studies, 10 µM Yoda1 was added to the cells and then immediately analyzed via flow cytometry. This allowed for a real-time analysis of calcium flux. Calcium flux was calculated as a measure of ratiometric fluorescence by dividing the Fluo-4 fluorescence by the Fura Red fluorescence (a higher ratiometric fluorescence would correspond with higher calcium influx).

To measure the Yoda1-induced calcium influx over longer periods of time, 200,000 cells were plated in 12-well plates for 24 h. Resveratrol-treated samples were incubated with 50 µM resveratrol for 1 h at 37 °C before plating. The cells were then treated with 10 µM Yoda1 for 24 h, then incubated with 1 µM Fluo-4 for 30 min at 37 °C. Cells were washed in calcium/magnesium-free HBSS, resuspended in HBSS with calcium and magnesium, then imaged using the green channel on an Olympus fluorescence microscope. Intracellular calcium was calculated by measuring the mean fluorescence intensity per cell and subtracting the background fluorescence.

### 5.10. Confocal Microscopy

SW480 and SW620 cells were seeded onto polystyrene cell-culture slides (Thermo Fisher Scientific, Waltham, MA, USA). Cells were allowed to grow for 24 h at 37 °C. The culture medium was replaced with serum-free media in the presence or absence of 50 µM resveratrol for 1 h at 37 °C. Cells were then washed and lipid rafts stained using the Vybrant Alexa Fluor 488 Lipid Raft Labeling Kit (Invitrogen, V34403). Briefly, cells were incubated with Alexa488-conjugated cholera toxin subunit B (CT-B) followed by an anti-CT-B antibody to crosslink CT-B labeled rafts. Slides were fixed for 15 min with 4% paraformaldehyde (PFA) (Electron Microscopy Sciences) in PBS (Gibco) and then permeabilized using 0.1% Triton X-100 (Millipore Sigma) in PBS at RT. Slides were blocked for 2 h with 5% goat serum (Thermo Fisher Scientific) and 5% bovine serum albumin (BSA, Sigma) in HBSS. Primary staining was conducted overnight at 4 °C with Piezo1 monoclonal antibody (Invitrogen clone 2–10) at a ratio of 1:100. Secondary staining was carried out with Alexa Fluor 555 goat anti-mouse IgG (H + L) (Invitrogen, A28180) for 45 min at RT (1:1000). Washes were conducted twice between each step for 5 min each using 0.02% Tween20 in PBS. Slides were assembled using 10 μL of Vectrashield antifade mounting media with DAPI (Vector Laboratories, Newark, CA, USA). Confocal imaging was performed using an LSM 880 (Carl Zeiss, Thornwood, NY, USA) with a 63×/1.40 Plan-Apochromat Oil, WD = 0.19 mm objective. At least five images were acquired per sample replicate. The Pearson’s correlation coefficient was calculated for each image using the JACoP plugin in ImageJ [56].

### 5.11. Flow Cytometry and FRET

Cells were cultured to 70% confluency upon collection and split into 250,000 cells per sample. Cells were fixed in 4% PFA in HBSS for 15 min at RT, then permeabilized with 0.1% Triton X-100 for 15 min with two HBSS washes between each step. The samples were blocked in 100 μL FACS buffer for 30 min at 4 °C, then stained with 1 μL AlexaFluor488-conjugated Piezo1 antibody (Novus Biologicals, NBP1-78446AF488) for 30 min at 4 °C. Samples were washed twice with HBSS and analyzed using a Guava easyCyte flow cytometer.

A FRET analysis via flow cytometry was performed as described above, with the added step of staining for lipid rafts. After blocking in FACS buffer, the cells were stained for LRs using the Vybrant Alexa Fluor 555 Lipid Raft Labeling Kit (Invitrogen, V34404). Piezo1 staining was then carried out as previously described. Donor quenching FRET efficiency was calculated using the following formula:$$E=1-\frac{{\mathrm{FI}}_{\mathrm{LR}+\mathrm{Piezo}}-{\;\mathrm{FI}}_{B}}{{\mathrm{FI}}_{\mathrm{Piezo}}-{\mathrm{FI}}_{B}},$$
where E is the FRET efficiency, FI_LR+Piezo_ is the mean fluorescence intensity of the double-stained LR/Piezo1 sample (acceptor + donor), FI_Piezo_ is the mean fluorescence intensity of the Piezo1-only stain (donor only), and FI_B_ is the fluorescence intensity of an unstained sample (background). The fluorescence intensity was recorded in the donor (Green-B) channel.

### 5.12. Computational Model of Cellular Apoptosis

The computational model was modified from a previously derived MATLAB-based model of Piezo1 sensitization of cancer cells to TRAIL-mediated apoptosis [19]. The simulations utilize ODEs to calculate changes in protein concentrations via the mass-action kinetics in the TRAIL-Piezo1-apoptosis pathway over time after treatment with TRAIL and various calcium-flux stimulants. The main modification to the model was the addition of a random, normally distributed Bcl-2 and XIAP-concentration generator for the random population analysis. Random–normal concentrations of cytosolic Bcl-2 (mean = 1.98 × 10^6^; median = 1.98 × 10^6^; standard deviation = 9.91 × 10^5^) and XIAP (mean = 1.01 × 10^5^; median = 1.01 × 10^5^; standard deviation = 9.91 × 10^4^) were generated, and the model was run using 50 ng/mL as the TRAIL treatment. Simulated cells were considered apoptotic once their calculated cPARP exceeded a concentration of 5 × 10^5^ molecules per cell. Cell viability was calculated as the percentage of cells which had not exceeded this apoptosis threshold. The biochemical reactions and their associated rate constants are listed in Appendix A. The non-zero initial conditions used in the model are displayed in Appendix A.

## Figures and Tables

**Figure 1 molecules-27-05430-f001:**
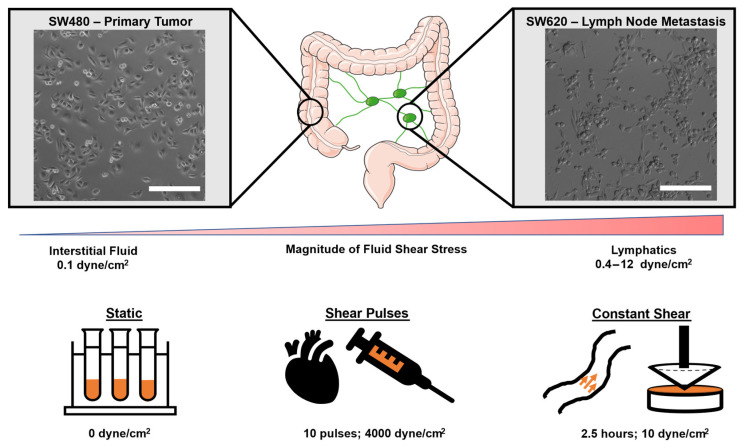
Isogenic colorectal cancer cellular model with different exposures to FSS in situ. FSS conditions were chosen to model flow conditions experienced by CTCs at brief intervals of turbulence (shear pulses) and average magnitudes and durations in the vasculature (constant shear).

**Figure 2 molecules-27-05430-f002:**
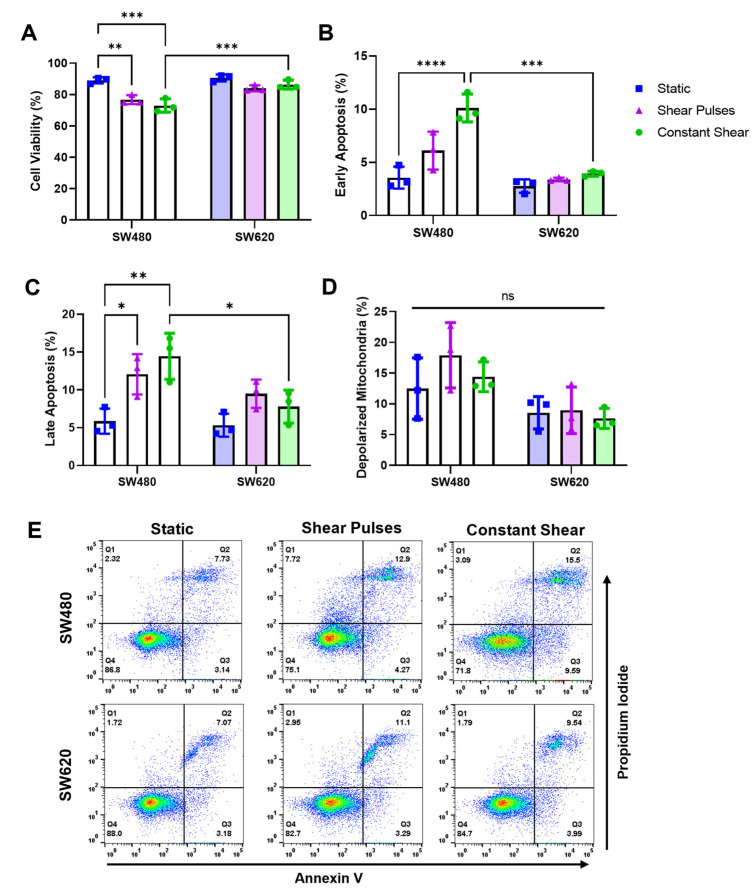
SW480 cells are more sensitive to FSS compared to SW620 cells. (**A**–**C**) Cell viability and apoptosis measured via Annexin V/PI staining 24 h after FSS treatments. (**D**) Percentage of cells with depolarized mitochondria measured via a JC-1 assay after FSS. N = 3, *ns =* not significant, * *p* < 0.05, ** *p* < 0.01, *** *p* < 0.001, ***** p* < 0.0001. (two-way ANOVA with multiple comparisons). Error bars represent mean ± SD. (**E**) Representative flow cytometry plots showing necrotic cells (Q1), late apoptotic cells (Q2), early apoptotic cells (Q3), and viable cells (Q4).

**Figure 3 molecules-27-05430-f003:**
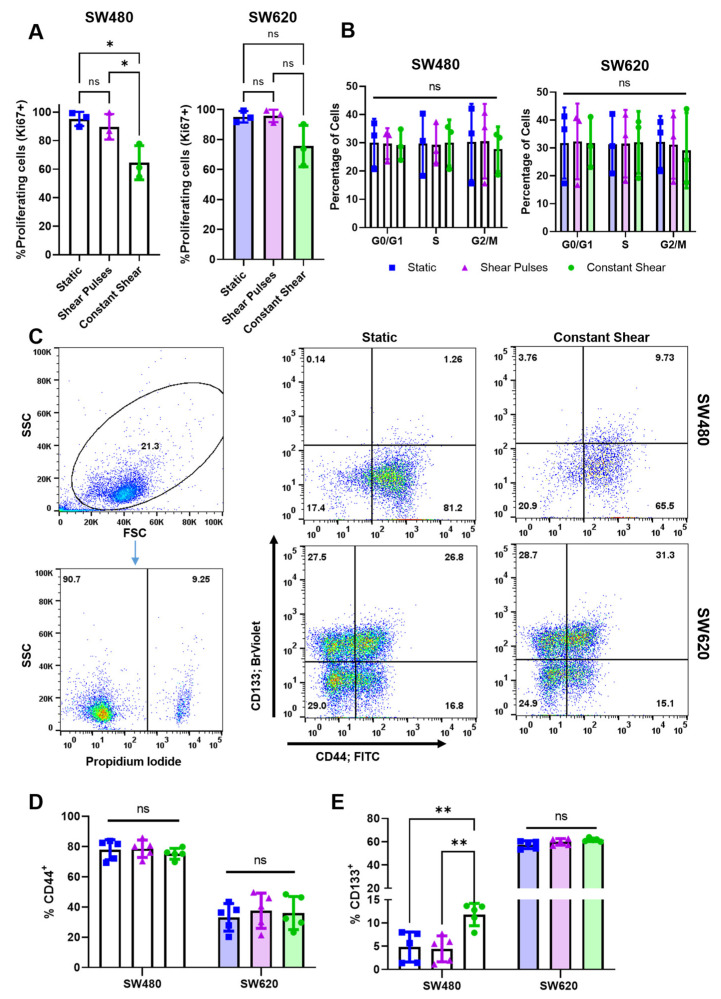
Constant shear stress decreases cell proliferation and increases CD133 expression in SW480 cells. (**A**) Percentage of proliferating cells 24 h post-FSS exposure, determined by the percentage of Ki67+ cells. N = 3, * *p* < 0.05 (one-way ANOVA with multiple comparisons). Error bars represent mean ± SD. (**B**) Cell-cycle distribution based on total DNA content from DAPI staining analyzed via flow cytometry. A Watson (Pragmatic) model was used in FlowJo to determine the percentage of cells in each stage of the cell cycle. N = 3. (**C**) Expression of CSC markers CD44 and CD133 post-FSS. Cells were gated from debris using forward and side scatter. PI+ cell populations were considered dead and excluded from analysis. Intact, PI- cells were appropriately compensated and gated for expression of CD44 and CD133 using single stained controls. (**D**,**E**) Quantification of CSC markers CD44 and CD133 for different conditions of FSS. N = 5, *ns =* not significant, ** *p* < 0.01 (two-way ANOVA with multiple comparisons). Error bars represent mean ± SD.

**Figure 4 molecules-27-05430-f004:**
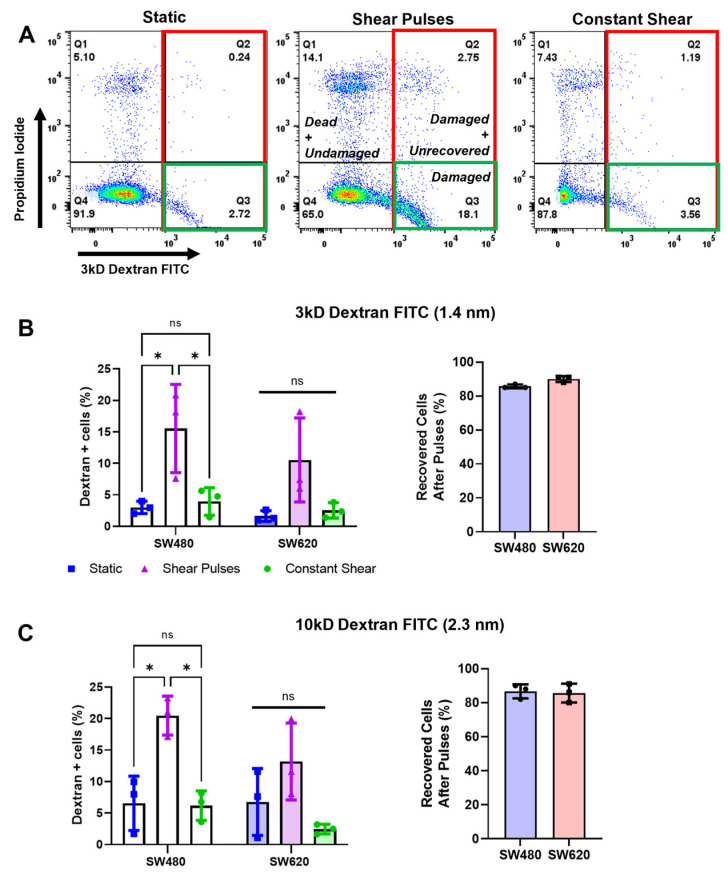
Shear pulses cause cell-membrane damage, indicated by membrane-pore formation, which is rapidly repaired. (**A**) Representative flow plots demonstrating enhanced dextran internalization after pulses of FSS. The percentages of damaged cells are shown in red and green (dextran+, Q3 and Q2), while the percentages of damaged cells that were unrecovered are shown in green (dextran+ and PI+, Q2 only). (**B**,**C**) Quantification of cell-membrane damage and repair for 3kD and 10kD MW dextran, respectively. Percentage of recovered cells was calculated by dividing the population of damaged and recovered cells (Dextran+/PI+, Q3) by the total number of damaged cells (Dextran+, Q2 + Q3). N = 3, *ns =* not significant, * *p* < 0.05 (two-way ANOVA with multiple comparisons). Error bars represent mean ± SD.

**Figure 5 molecules-27-05430-f005:**
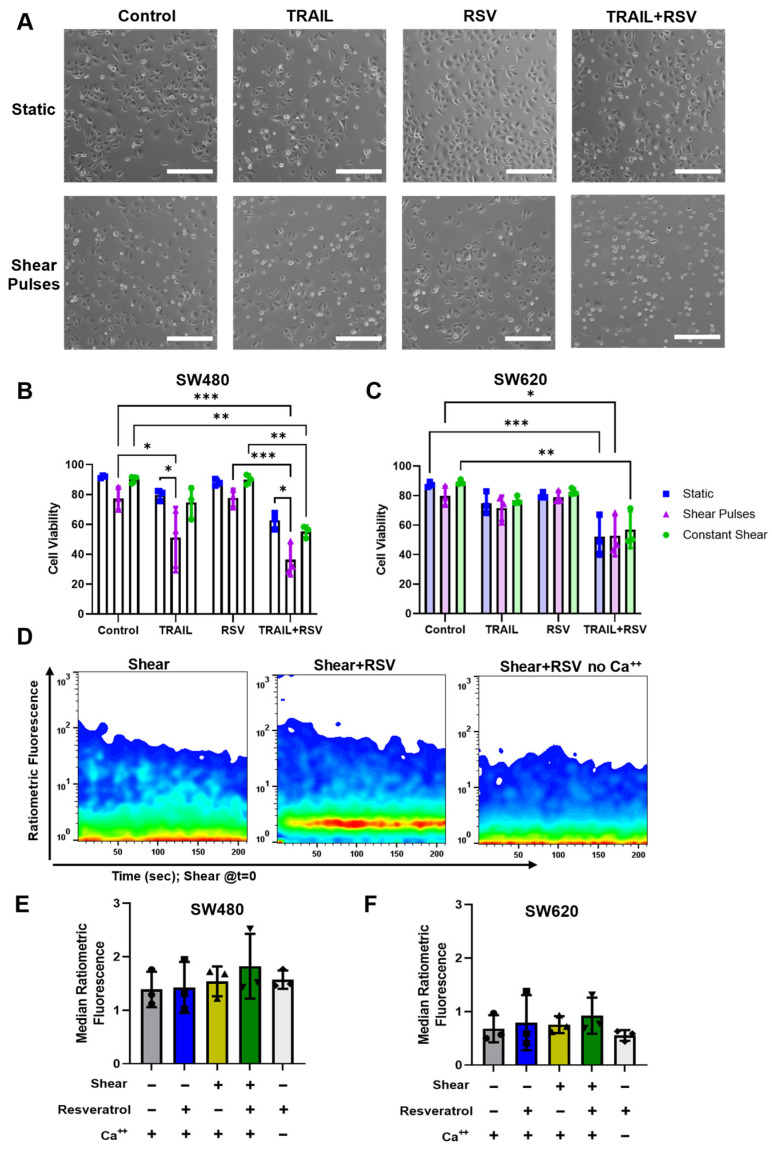
Shear pulses and resveratrol sensitize SW480 cells to TRAIL individually but have minimal synergistic effects. (**A**) Phase contrast images of SW480 cells 24 h after FSS treatment with 50 ng/mL TRAIL and 50 µM resveratrol. Samples exposed to shear pulses and treated with TRAIL + RSV displayed an increased number of apoptotic cells, as visualized by a rounded and shrunken morphology. Scale bar = 100 µm. (**B**,**C**) The viabilities of SW480 and SW620 cells measured via AnnexinV/PI staining following FSS treatments with or without 50 ng/mL TRAIL and 50 µM resveratrol. N = 3, * *p* < 0.05, ** *p* < 0.01, *** *p* < 0.001 (two-way ANOVA with multiple comparisons). Error bars represent mean ± SD. (**D**) Ratiometric fluorescence of Fluo-4/FuraR as a metric of intracellular calcium concentration in real time. Cells were sheared with one pulse at *t* = 0 then immediately analyzed via flow cytometry. (**E**,**F**) Insignificant changes in median fluorescence intensity of ratiometric calcium fluorescence for SW480 and SW620 cells, respectively. Median fluorescence was calculated over the course of the first 200 sec. N = 3 (two-way ANOVA with multiple comparisons). Error bars represent mean ± SD.

**Figure 6 molecules-27-05430-f006:**
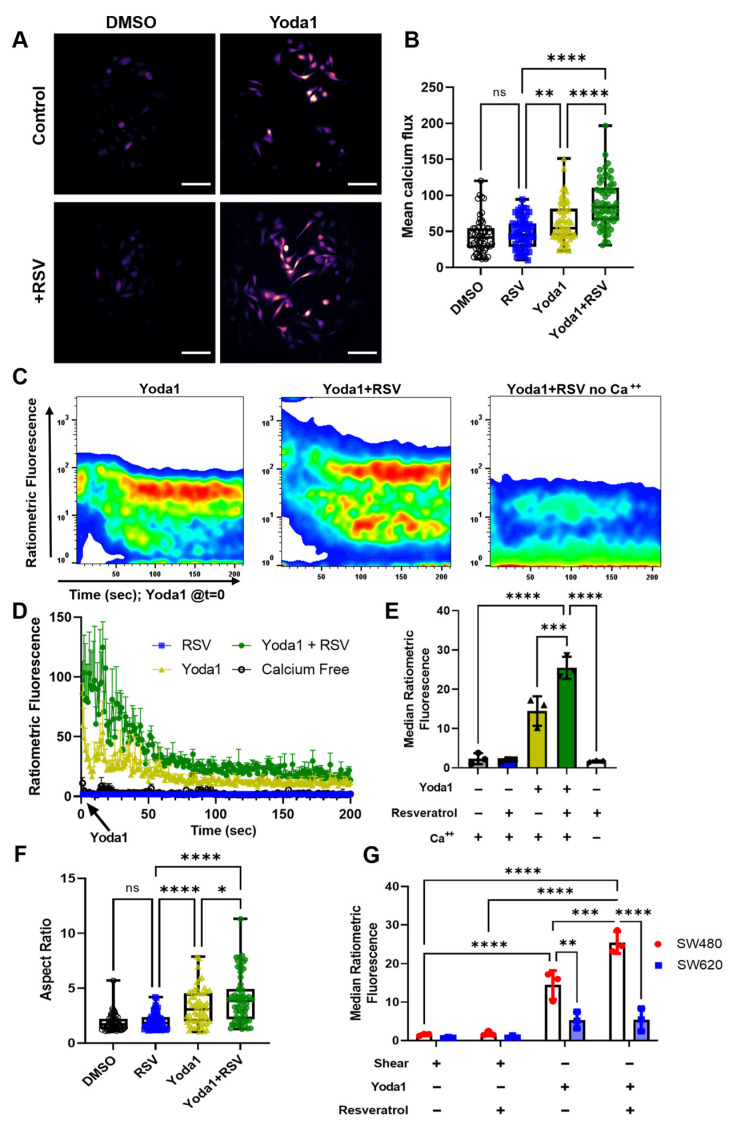
Resveratrol increases calcium influx from Yoda1 in the more mechanosensitive SW480 cells. (**A**) Calcium imaging of Fluor-4 fluorescence after 24 h treatment of 10 µM Yoda1. Scale bar = 100 µm. (**B**) Quantification of mean calcium flux per cell after 24 h Yoda1 treatment. N = 2 (60 cells per condition), ** *p* < 0.01, **** *p* < 0.0001. (**C**) Representative time-course flow cytometry plots showing instantaneous changes in SW480 calcium influx as measured by the ratiometric fluorescence of Fluo-4/FuraR. Yoda1 (10 µM) added at *t* = 0. (**D**) Median ratiometric fluorescence as a function of time in SW480 cells. N = 3. Error bars represent mean + SEM. (**E**) Average ratiometric calcium fluorescence over 200 s after Yoda1 treatment. N = 3, *** *p* < 0.001, **** *p* < 0.0001 (one-way ANOVA with multiple comparisons). Error bars represent mean ± SD. (**F**) SW480 aspect ratio after 24 h treatment of Yoda1 (10 µM) and resveratrol (50 µM). N = 2 (60 cells per condition), * *p* < 0.05, **** *p* < 0.0001. (**G**) Calcium influx comparing mechanical shear with Yoda1 activation of Piezo1 in SW480 and SW620 cells. N = 3, *ns =* not significant, ** *p* < 0.01, *** *p* < 0.001, **** *p* < 0.0001 (two-way ANOVA with multiple comparisons). Error bars represent mean ± SD.

**Figure 7 molecules-27-05430-f007:**
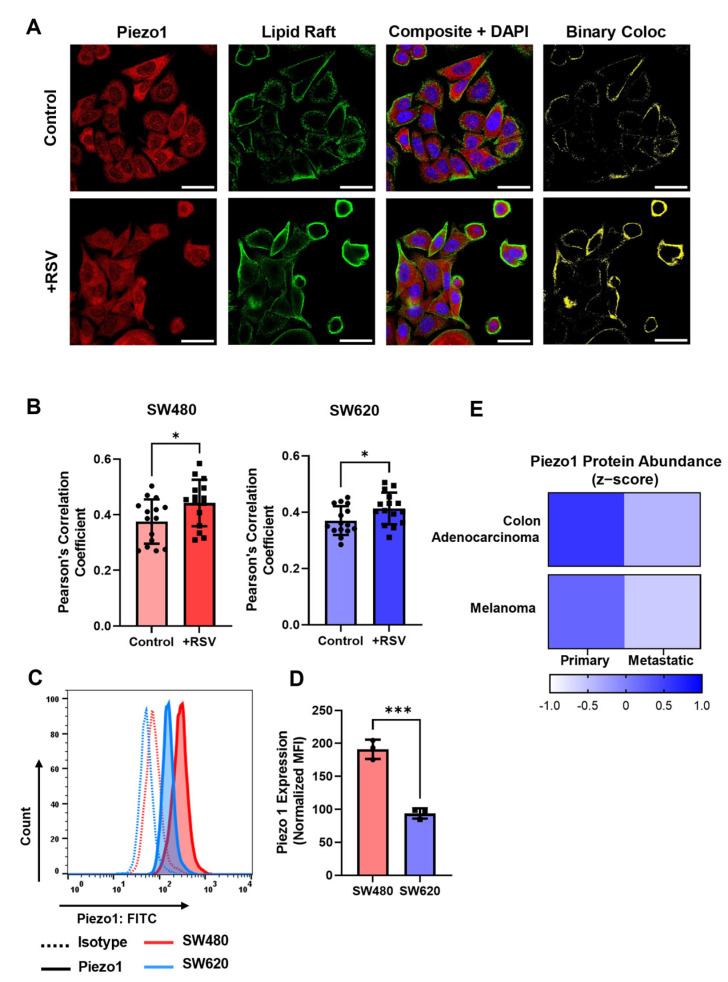
Resveratrol increases LR/Piezo1 colocalization, and SW480 cells have higher Piezo1 expression. (**A**) Confocal microscopy images of Piezo1 (red), lipid rafts (green), DAPI (blue), and colocalization events between Piezo1 and LR (yellow). Scale bar = 30 µm. (**B**) Pearson’s correlation coefficient between LR/Piezo1 was calculated using the JACoP plugin in ImageJ. N = 3 (*n* = 15 images), * *p* < 0.05 (two-tailed unpaired t-test). For all plots, error bars represent mean ± SD. (**C**) Expression of Piezo1 (solid line) and isotype controls (dashed lines) in SW480 (red) and SW620 (blue) cells. (**D**) Median fluorescence intensity of Piezo1 normalized to the respective isotype control. N = 3, *** *p* < 0.001 (two-tailed unpaired *t*-test). (**E**) Protein abundance data (z-scores) from the Broad Cancer Cell Line Encyclopedia, downloaded from cBioPortal. Primary tumor and lymph node metastases from the same patient with colon adenocarcinoma (SW480 and SW620) and melanoma (WM115 and WM2664). Darker blue represents a higher z-score and higher Piezo1 abundance.

**Figure 8 molecules-27-05430-f008:**
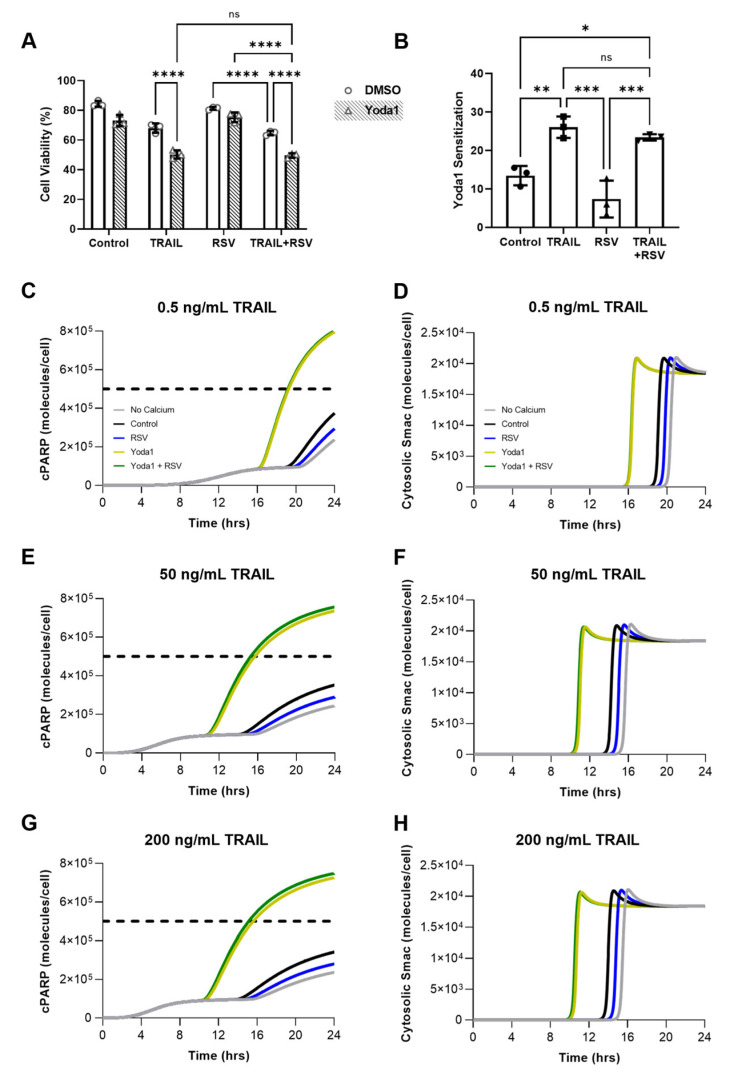
Resveratrol has a minimal effect on the TRAIL sensitizing ability of Yoda1. (**A**) SW480 cells treated for 24 h with 10 µM Yoda1 and 50 ng/mL TRAIL with or without 50 µM pretreatment with resveratrol. N = 3, **** *p* < 0.0001 (two-way ANOVA with multiple comparisons). Error bars represent mean ± SD. (**B**) Yoda1 sensitization from cell viabilities shown in (**A**). N = 3, *ns =* not significant, * *p* < 0.05, ** *p* < 0.01, *** *p* < 0.001 (one-way ANOVA with multiple comparisons). Error bars represent mean ± SD. (**C**–**H**) Computational model of apoptosis for 0.5 ng/mL, 50 ng/mL and 200 ng/mL TRAIL measuring concentrations of cleaved PARP and cytosolic Smac over time. In vitro calcium concentrations were used and normalized to Yoda1 (1 µM). Cells were considered apoptotic at cPARP concentrations exceeding 5 × 10^5^ molecules/cell.

**Figure 9 molecules-27-05430-f009:**
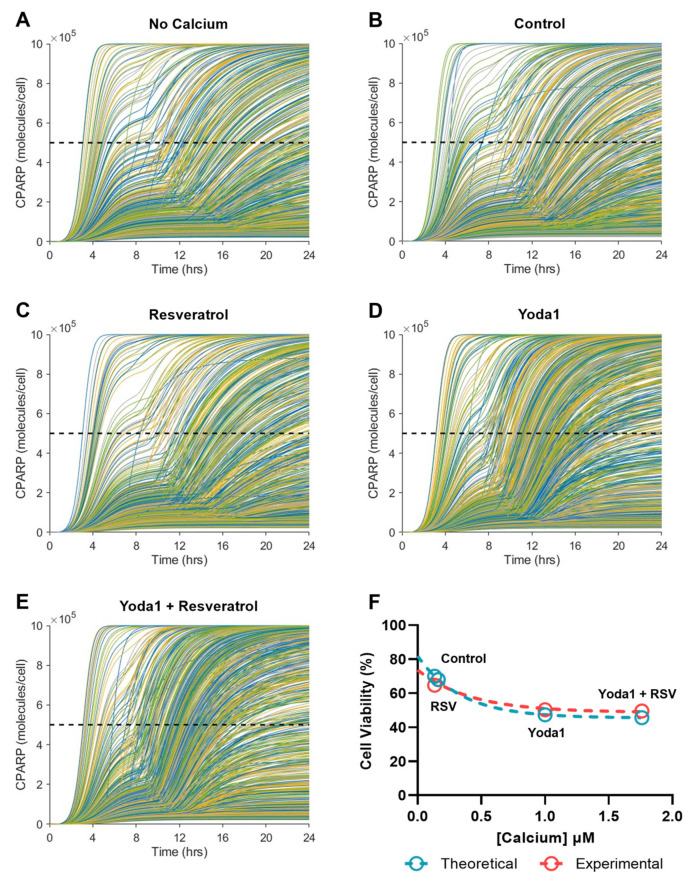
Computational modeling and experimental data demonstrate the saturating effects of increased calcium on cell apoptosis. (**A**–**E**) Simulation of heterogeneous cell populations with randomly generated, but normally distributed, cytosolic Bcl-2 and XIAP concentrations. Each line represents one cell with a randomized concentration of each protein, and each simulation was run for 1000 cells. A cPARP concentration of 5 × 10^5^ molecules/cell was used as the apoptotic threshold for each cell (cells above this line after 24 h were considered apoptotic). Cell viability was determined as the percentage of cells below this concentration after 24 h. (**F**) Theoretical (blue) and experimental (red) cell viabilities as a function of measured calcium influx (normalized to Yoda1) (nonlinear regression, one phase decay).

## Data Availability

The codes used in this study are provided in the following GitHub repository https://github.com/marialopezcavestany/cancer-piezo1-resveratrol accessed on 8 August 2022. The authors request that these programs not be modified or distributed without attribution to this published work. Data within the manuscript will be shared upon request by the corresponding author Michael King (mike.king@vanderbilt.edu).

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
