# Peer review of "Piezo1 Mechano-Activation Is Augmented by Resveratrol and Differs between Colorectal Cancer Cells of Primary and Metastatic Origin"

_molecules, 2022, doi:10.3390/molecules27175430_

Round 1

Reviewer 1 Report

In the proposed manuscript, titled “Piezo1 mechano-activation is augmented by resveratrol and differs between colorectal cancer cells of primary and metastatic origin”, the authors investigated in depth the impact of Fluid Shear Stress (FSS) on colorectal cancer cell proliferation, apoptosis, membrane damages, calcium influx, and therapeutic sensitization. For this purpose, they submitted two cell lines - SW480 (the primary tumor) and SW620 cells (lymph node metastasis), derived from the same patient- to different mechanical stimuli (Static, shear pulses, and constant shear). Then, the authors investigated cell membrane damage, cell proliferation, and the expression of CD113 and CD44. In addition, they studied the role of mechanosensitive ion channels, prior and after the pre-treatment with resveratrol.

The proposed manuscript might be very interesting and relevant to the pharmaceutics, biotechnological, and medical community; however, it requires major changes before publication. There are some relevant formal and technical issues that must be addressed:

1.      The manuscript is well written, and no relevant grammar errors have been found. Anyway, the manuscript is full of abbreviations and acronyms, some of which are commonly used in numerous scientific articles while others are more specific and less common. Usually, the use of acronyms should facilitate the reader but, not in this case. In my opinion, it is mandatory to insert a list of acronyms at the beginning of the manuscript.

2.      The use of the RGB (red/green/blue) colors in the schemes and graphics reported in the manuscript is unreadable for color-blind people and as such no longer acceptable in today’s publishing. The authors should consider popular scientific coloring guides (such as https://journals.plos.org/plosone/article?id=10.1371/journal.pone.0199239 ) and must change the color palette of the schemes.

3.      The authors must revise all histograms for several reasons: The bars are too small (Fig 3a), 5e) 8a)); the errors bars have the same color of the bars of the histogram and it is impossible to distinguish them; use different colors for the dots and the histogram bars.

4.      the conclusions section is completely missing. The authors must report it in a proper section (section n° 5).

Author Response

Molecules-1832650

Piezo1 mechano-activation is augmented by resveratrol and differs between colorectal cancer cells of primary and metastatic origin

Response to Reviewers

We thank the reviewers for their diligence in reviewing our manuscript and for their helpful comments. All changes to the manuscript are marked up using Track Changes. Please find a detailed response to each reviewer comment below.

Reviewer 1:

  1. The manuscript is well written, and no relevant grammar errors have been found. Anyway, the manuscript is full of abbreviations and acronyms, some of which are commonly used in numerous scientific articles while others are more specific and less common. Usually, the use of acronyms should facilitate the reader but, not in this case. In my opinion, it is mandatory to insert a list of acronyms at the beginning of the manuscript.

We have inserted, before the introduction, a complete list of all abbreviations and acronyms used in the manuscript.

  1. The use of the RGB (red/green/blue) colors in the schemes and graphics reported in the manuscript is unreadable for color-blind people and as such no longer acceptable in today’s publishing. The authors should consider popular scientific coloring guides (such as https://journals.plos.org/plosone/article?id=10.1371/journal.pone.0199239 ) and must change the color palette of the schemes.

We agree that the use of red/green/blue color schemes for the indication of FSS treatments was an oversight on our part and not inclusive for colorblind readers. We have changed the schemes on all figures, changing the red scheme for “shear pulses” to magenta (new color scheme being blue, magenta, green) to make it appropriate for users with deuteranopia and/or protanopia (https://www.ascb.org/science-news/how-to-make-scientific-figures-accessible-to-readers-with-color-blindness/). We also darkened the tone of green in the “Yoda1+RSV” condition (i.e. Figure 6D) to make it more distinguishable from the “Yoda1” condition in yellow. Additionally, even in grayscale, all conditions can be identified from different symbol shapes (i.e square for static, triangle for shear pulses, and circle for constant shear).  

  1. The authors must revise all histograms for several reasons: The bars are too small (Fig 3a), 5e) 8a)); the errors bars have the same color of the bars of the histogram and it is impossible to distinguish them; use different colors for the dots and the histogram bars.

To better read and interpret the graphs, we have increased the error bar thickness on all graphs and changed its color to be different from the underlying histogram bars that display the average. We have also made the bar outlines black, with the error bars and individual points colored to help distinguish these, especially in instances where the bars are smaller and the points overlap with the mean. For example, in figure 8a, the error bars were moved to lay “overtop” of the points for better visualization, and the individual data points were changed to a lighter color.   

  1. “the conclusions section is completely missing. The authors must report it in a proper section (section n° 5).”

We have added a conclusion section following the discussion where we highlight the main findings of the paper and speak on the clinical significance of the results.

Reviewer 2 Report

In this article, J.D. Greenlee and colleagues have evaluated the potential differences in the mechano-activation of CRC cells in an isogenic cellular model, derived from the primary tumour and the metastasis of the same CRC patient, investigating the response to different patterns of fluid shear stress (FSS) and treatment with Resveratrol.

The manuscript is well written and the parameters involved in the research are reported in detail.

The effects of static, shear pulses and constant shear and the response to Resveratrol have been properly described in the results and clearly addressed in the figures. Furthermore, both tables and figures in the supplementary data are clear and complete the information of the main text.

Minor comments:

In chapter 2.8, please doublecheck the citation of Fig.8C, 8D, 8E and 8F.

In addition, it could be useful to define the main results, and mostly the clinical implications of the effects observed, as conclusions of the manuscript and to better feature them in the abstract.

Author Response

Molecules-1832650

Piezo1 mechano-activation is augmented by resveratrol and differs between colorectal cancer cells of primary and metastatic origin

Response to Reviewers

We thank the reviewers for their diligence in reviewing our manuscript and for their helpful comments. All changes to the manuscript are marked up using Track Changes. Please find a detailed response to each reviewer comment below.

Reviewer 2:

  1. In chapter 2.8, please doublecheck the citation of Fig.8C, 8D, 8E and 8F.

Thank you for this observation, Figures 8C, 8D, 8E and 8F had been referenced incorrectly in the text. We have edited the text to properly cite these to their corresponding figure panels.

  1. In addition, it could be useful to define the main results, and mostly the clinical implications of the effects observed, as conclusions of the manuscript and to better feature them in the abstract.

We have added a conclusion section that we feel better highlights the main results of the paper. Additionally, we use this section to speak on the clinical significance of our results and how these findings may improve treatment decisions moving forward. We also slightly modified the abstract to better speak on the translational significance of these findings.   

Round 2

Reviewer 1 Report

I very much appreciated the changes made to the manuscript.

The conclusion section is well written In my opinion, the changes have greatly improved the quality of the article.